# Staphylococcal Scalded Skin Syndrome (SSSS) in a Preterm Infant: Clinical Presentation and Role of Next-Generation Sequencing in Toxin Gene Identification—A Case Report

**DOI:** 10.3390/antibiotics14101017

**Published:** 2025-10-13

**Authors:** Giovanni Lorenzin, Maddalena Carlin, Claudio Scarparo, Mariachiara Cardellini, Francesca Tota, Aldo Naselli

**Affiliations:** 1Microbiology and Virology Unit, S. Chiara Hospital, 38122 Trento, Largo Medaglie d’oro, Italy; giovanni.lorenzin@apss.tn.it (G.L.); claudio.scarparo@apss.tn.it (C.S.); 2Department of Cellular, Computational and Integrative Biology CIBIO, University of Trento, 38123 Trento, Largo Medaglie d’oro, Italy; maddalena.carlin@studenti.unitn.it; 3Neonatal Intensive Care Unit, S. Chiara Hospital, 38122 Trento, Largo Medaglie d’oro, Italy; mariachiara.cardellini@apss.tn.it (M.C.); francesca.tota@apss.tn.it (F.T.)

**Keywords:** staphylococcal scalded skin syndrome (SSSS), *Staphylococcus aureus*, exfoliative toxins, methicillin resistance, whole-genome sequencing (WGS), virulence factors, catheter-related infections

## Abstract

Staphylococcal scalded skin syndrome (SSSS) is a rare, toxin-mediated dermatosis caused by exfoliative toxin–producing *Staphylococcus aureus* strains, with neonates and preterm infants being particularly vulnerable due to immature immunity and reduced toxin clearance. We report the case of a male preterm infant, born at 24 weeks of gestation, who presented at the age of one month with fever and later developed widespread erythema, flaccid bullae, and periorificial desquamation. Methicillin-sensitive *S. aureus* (MSSA) was isolated from blood, catheter, and auricular swabs. Whole-genome sequencing revealed sequence type ST121 carrying both *eta* and *etb* genes, confirming the dual-toxin profile associated with severe disease. The infant improved with targeted intravenous oxacillin following catheter removal. A subsequent nasal swab identified a methicillin-resistant *S. aureus* (MRSA) ST30 strain lacking exfoliative toxins, consistent with asymptomatic colonization. This case underscores the importance of integrating advanced molecular diagnostics such as next-generation sequencing into the management of neonatal SSSS, enabling precise identification of virulence factors and resistance genes. Literature also highlights the global epidemiology of SSSS, diversity of *S. aureus* toxin genes, and value of genomic surveillance in neonatal care; our case aligns with reports of ST121 strains carrying both ETA and ETB, where the dual-toxin profile drives rapid onset, extensive skin disease, and good outcomes with prompt therapy.

## 1. Introduction

Staphylococcal scalded skin syndrome (SSSS) is a rare but severe exfoliative skin disease within the spectrum of skin and soft tissue infections caused by *Staphylococcus aureus* in neonates [1]. The syndrome is caused by specific *Staphylococcus aureus* strains that produce exfoliative toxins A and B (ETA and ETB), which are potent epidermolytic exotoxins that disrupt epidermal integrity by cleaving desmoglein-1 (Dsg1), a cadherin protein responsible for keratinocyte adhesion within the granular layer of the epidermis [2,3,4]. ETA is encoded by the *eta* gene located on a temperate bacteriophage (φETA), whereas ETB is encoded by the *etb* gene carried on a large plasmid [5]. Upon systemic hematogenous dissemination, these toxins induce intraepidermal cleavage, resulting in widespread erythema, flaccid bullae formation, and subsequent skin desquamation [5]. Although exfoliative toxin production occurs in only approximately 5% of *S. aureus* isolates—with ETA present in about 89% of toxin-producing strains, ETB in 4%, and both toxins concurrently in approximately 7%—neonates and infants exhibit heightened susceptibility [6]. This vulnerability is attributable to their limited neutralizing antibody capacity and immature renal clearance, which impairs toxin elimination. Clinically, affected infants typically present with tender erythema localized to flexural and periorificial regions, a positive Nikolsky’s sign (denoting the detachment of the superficial epidermis in response to gentle lateral pressure), and systemic symptoms including fever, irritability, and feeding difficulties [4,7].

Neonates and infants with SSSS or other severe *S. aureus* infections requiring hospitalization are frequently managed with central venous catheters (CVCs) for administration of parenteral nutrition, intravenous antibiotics, or supportive care. The presence of a CVC in these vulnerable patients significantly elevates the risk of developing catheter-related bloodstream infections (CRBSI). *S. aureus* is among the leading pathogens responsible for CRBSI in pediatric and neonatal settings, with infections often occurring through colonization of catheter hubs or hematogenous seeding from cutaneous or mucosal sites already compromised in SSSS [8,9].

Staphylococcal scalded skin syndrome may manifest across all age groups; however, its prevalence is highest among children younger than six years, with cases reported as early as the neonatal period [7,10]. According to findings from research conducted in the Czech Republic, there are around 25 incidences for every 100,000 infants under one year old [11].

While diagnosis has traditionally been based on clinical features supported by bacterial culture from primary infection sites, advances in molecular diagnostics now allow for rapid and precise pathogen characterization. Next-generation sequencing (NGS) has emerged as a valuable tool in clinical microbiology, enabling simultaneous identification of bacterial strains, detection of virulence factors such as toxin ETA and ETB, and assessment of antimicrobial resistance genes [12]. Early application of NGS in neonatal sepsis and toxin-mediated syndromes can enhance diagnostic accuracy, inform targeted therapy, and aid in epidemiological surveillance, particularly in cases involving uncommon or highly virulent clones like *S. aureus* sequence type 121 (ST121).

Here, we present the case of a preterm infant with SSSS caused by *S. aureus* ST121, in which NGS was employed to confirm the presence of exfoliative toxin genes. This report highlights the clinical course, microbiological findings, and the potential role of genomic technologies in improving the diagnosis and management of toxin-mediated staphylococcal diseases in vulnerable neonatal populations.

## 2. Case Presentation

A male infant, born extremely premature at 24 weeks of gestation (birth weight 880 g), presented at a chronological age of 4.5 months (corrected age 1 month) with fever (peak 38.3 °C), mild respiratory symptoms, and decreased feeding. The infant had a complex past medical history notable for necrotizing enterocolitis (NEC) requiring enterostomy, with ongoing dependence on nighttime parenteral nutrition delivered via a CVC, and was under pediatric and surgical follow-up.

On general physical examination, the baby was alert, irritable, with tone and reactivity appropriate for his age, pink-pale skin, and no respiratory distress or hemodynamic compromise, presenting only rough upper airway noises. Examination revealed stable stomas and peristomal skin, normal abdominal and cardiac findings.

Initial laboratory investigations included negative syndromic panel from nasopharyngeal swabs for common respiratory viruses (QIAstat-Dx Respratory SARS-CoV-2 Panel, Quiagen, Hilden Germany), but subsequent testing via Filmarray Pneumoplus (BioFire Diagnostics, bioMérieux, Salt Lake City, UT, USA) identified rhinovirus/enterovirus, as well as *Escherichia coli* and *Serratia marcescens* genomes in the nasopharynx.

Restrained for fever, on the 4th day of hospitalization, the infant developed a diffuse erythematous rash with flaccid bullae and periorificial desquamation, especially in the facial area (Figure 1), with Nikolsky sign positive. Blood and catheter cultures were processed on the VIRTUO automated blood culture system (bioMérieux, Marcy-l’Étoile, France). FilmArray BCID panel (BioFire Diagnostics, bioMérieux, USA) performed on positive bottles detected *Staphylococcus aureus* and did not detect *mecA* or SCCmec targets. Subcultures on agar plates (Kima s.r.l., Padua, Italy) were identified using MALDI-TOF mass spectrometry (Vitek MS, Marcy-l’Étoile, bioMérieux). Antimicrobial susceptibility testing was performed by broth microdilution on the MicroScan WalkAway system (Beckman Coulter, Brea, CA, USA) and interpreted using EUCAST clinical breakpoints. The isolate was phenotypically susceptible to oxacillin/cefoxitin and therefore classified as methicillin-susceptible *Staphylococcus aureus* (MSSA). Nasal swabs for MRSA screening were performed weekly as part of the routine surveillance. In light of the positive blood cultures, skin swabs or nasal swabs for MSSA were not performed.

Dermatological evaluation confirmed staphylococcal scalded skin syndrome (SSSS). The microbiology unit planned to sequence the bacterium to confirm the production of exfoliative toxins. Infectiological assessment attributed the clinical condition to a CVC-related MSSA sepsis (catheter-related bloodstream infection, CRBSI) (Figure 2) complicated by SSSS. Vancomycin therapy was initiated at a dosage of 60 mg/kg/day, divided into four daily administrations, from the day of clinical onset until the availability of the phenotypic antibiotic susceptibility results from blood cultures. After the AST from the microbiology, therapy was de-escalated to oxacillin at a dosage of 200 mg/kg/day, also divided into four daily administrations.

The CVC was removed and replaced on day 12 due to catheter contamination. Skin lesions regressed, with minimal residual erythema.

The patient maintained adequate growth and nutritional intake through cycles of parenteral and enteral feeds, with close monitoring and tailored stoma and skin care. Prior to discharge (hospital day 24), the patient’s condition had normalized, with stabilized parameters, adequate weight gain (weight at discharge 3800 g), and satisfactory cardiopulmonary, abdominal, and stoma function.

### 2.1. DNA Extraction and Whole Genome Sequencing

The *S. aureus* isolate recovered from the blood culture underwent whole-genome sequencing. The isolate was cultured on Tryptone Soy Agar with Sheep Blood. Single colonies were transferred to Tryptone Soy Broth and incubated aerobically overnight at 37 °C.

Bacterial pellets were enzymatically pre-treated with 5 μL mutanolysin and 8 μL lysostaphin for 1 h at 37 °C, followed by 300 μL of lysis buffer and 30 μL proteinase K at 56 °C for 1 h.

Genomic DNA was extracted using the Maxwell^®^ RSC Pathogen Total Nucleic Acid Kit on the Maxwell^®^ RSC automated instrument (Promega Corporation, Madison, WI, USA), following the manufacturer’s protocol. DNA concentration was measured with the Qubit™ 4 Fluorometer using the Qubit™ dsDNA HS Assay Kit (Thermo Fisher, Waltham, MA, USA).

### 2.2. Library Preparation and Sequencing

Genomic DNA libraries were prepared using the Illumina DNA PCR-Free Library Preparation Kit, which enables the construction of high-quality sequencing libraries without the use of PCR amplification, thereby minimizing sequence bias and preserving genome representation. Pooled libraries were quantified via Qubit™ ssDNA Assay Kit and sequenced (2 × 150 bp) on an Illumina MiSeq Dx platform with MiSeq Reagent Kit v2 (300 cycles) and custom primers [13].

### 2.3. Bioinformatics Analysis

Reads were assembled de novo into contigs using Unicycler v0.5.0. A set of tools was used to determine the MLST (https://pubmlst.org/), the resistance was tested with ABRICATE NCBI, replicons with Plasmidfinder v2.1.6, and virulence factors using ABRICATE with VFDB. The genome was annotated using Prokka v1.14.6. and manually curated. Plasmids were identified using mobSUITE v3.1.9, pathogenic islands with islandwiewer v4.0.0, and phages using phaster v2.2.16.

## 3. Results

The genotype of the *S. aureus* isolate revealed a sequence type ST121, a globally disseminated and hypervirulent clone known to carry exfoliative toxin genes. The virulence factor profile of the isolate, associated with our case of staphylococcal scalded skin syndrome (SSSS), reveals a comprehensive arsenal of genes supporting its pathogenic potential in neonatal skin infections. Genetic analysis of the isolate revealed the presence of the *eta* and *etb* genes encoding the exfoliative toxins ETA and ETB, respectively, confirming the strain’s capacity to induce epidermal splitting via desmoglein--1 cleavage—central to the pathogenesis of SSSS. This dual-toxin profile is consistent with more severe clinical presentations and widespread desquamation [2,4,9]. Additionally, the strain possesses bi-component leukocidin genes *lukED* and gamma-hemolysin *hlgABC*, which has been recognized to contribute to immune evasion and cytotoxicity against host leukocytes [9].

The presence of delta-hemolysin, serine proteases (SplA and SplB), and aureolysin, a zinc metalloproteinase, further enhances the bacterium’s ability to disrupt epithelial barriers and modulate the host immune response. Multiple enterotoxins (types I, M, N, U, Y, Z, and X) were also identified, indicating a potential for superantigenic activity that could exacerbate systemic inflammation. Importantly *icaC* gene, part of the intercellular adhesin gene (*icaABCD*) locus, responsible for polysaccharide intercellular adhesin biosynthesis was detected, indicating biofilm-forming capability, which has been recognized a key factor in persistent colonization [9].

Resistance genes were widespread, including fosfomycin resistance (*fosB*/*fosD*), beta-lactamase production (*blaZ*), tetracycline efflux [*tet(38)*], and multiple multidrug efflux pumps (*mepA*, *lmrS*). The presence of these mechanisms highlights the strain’s ability to resist diverse antibiotic classes, posing a significant challenge for effective therapeutic management.

### Antibiotic Susceptibility Testing

Although genetic analysis reveals the presence of some resistance determinants, the antimicrobial susceptibility profile (Table 1) suggests that the Staphylococcus aureus isolate is a hypersensitive isolate, showing sensitivity (S) or high exposure sensitivity (I) to all antimicrobial agents tested. Antimicrobial susceptibility testing was performed by broth microdilution on the MicroScan WalkAway system (Beckman Coulter, Brea, CA, USA) and interpreted using EUCAST clinical breakpoints (Table 1).

One month after the acute episode of staphylococcal scalded skin syndrome (SSSS), the infant was found to be colonized with a methicillin-resistant *Staphylococcus aureus* (MRSA) strain, detected from routine surveillance nasal swab for MRSA colonization. Whole-genome sequencing of this isolate revealed that it belonged to sequence type ST30, carrying the SCCmec type IV element—a lineage distinct from the previously identified ST121 SSSS-causing strain. Importantly, this MRSA isolate did not produce exfoliative toxins A or B, the essential virulence factors responsible for epidermal cleavage in SSSS, which explains the absence of clinical symptoms during this colonization event.

Despite lacking the key toxins involved in SSSS pathogenesis, this ST30 strain harbored a robust repertoire of other virulence and resistance genes. Among them, the presence of the *mecA* gene encoding PBP2a confirms resistance to beta-lactams, supported by associated regulatory genes (*blaR1*, *mecR1*, and *blaZ*) and efflux systems such as *Tet*(*38*) and *MepA*. The isolate possessed TSST-1 toxin (toxic shock syndrome toxin), several staphylococcal enterotoxins (I, M, U), and SCIN-A (a complement inhibitor), suggesting potential for systemic disease.

Additional features, including gamma- and delta-hemolysins, zinc metalloproteinase aureolysin, serine protease SplE, and staphylokinase, enhance tissue invasion, immune evasion, and bacterial dissemination. The presence of *icaC* gene, linked to biofilm production, supports the strain’s ability to persist on mucosal surfaces. However, the lack of exfoliative toxins and the absence of disease signs suggest that this strain acted as a colonizer rather than an invasive pathogen.

## 4. Discussion

This case presents a rare yet instructive instance of SSSS in an extremely premature infant, complicated by catheter-related MSSA sepsis, and highlights the critical role of whole-genome sequencing (WGS) in confirming the presence of exfoliative toxin genes. Notably, the ST121 MSSA strain harbored both ETA and ETB toxins, concordant with literature emphasizing severe SSSS presentations from dual-toxin producers [14].

Premature infants, especially those born <28 weeks, face compounded vulnerability due to immature skin barriers, underdeveloped immune systems with low transplacental IgG due to the diminished transplacental antibody transfer that normally occurs in the third trimester, and frequent reliance on invasive devices such as central lines [15,16]. In addition, impaired renal clearance in premature infants reduces the elimination of circulating toxins, prolonging exposure and amplifying cutaneous injury. Another biologic factor may be the developmental variability in desmoglein (*dsg*) expression—neonates express *DSG-3* in superficial epidermal layers, which is unaffected by exfoliative toxins A and B, potentially offering partial protection compared to adults; however, the immaturity of skin barriers in preterm infants may offset any such protection, leaving them highly susceptible to widespread skin involvement [3]. Prior case reports have documented similar severe presentations: for instance, Rieger-Fackeldey et al. (2002) [14] described a 25-week preterm infant (364 g) who suffered recurrent SSSS from an ETA/ETB-positive MSSA strain, with evidence of staff-mediated transmission, underscoring the infection control challenges in NICUs. Hütten et al. reported SSSS as an initial sign of sepsis by an ETA/ETB producer in a very low birth weight preterm [17]. Methicillin sensitive Staphylococcus aureus strains, causative agent of SSSS, usually are community-acquired infections and are normally not hospital acquired infections. These align with our case both clinically and microbiologically. Supporting this, no other infants in the NICU presented with similar symptoms. For this type of pathogen pre-hospitalization screening is not feasible because it would require surveillance through whole genome sequencing (WGS), and in this case the criteria for an outbreak are not met.

Genomic epidemiology studies have shown global spread of ET-producing *S. aureus* lineages, with ETA typically phage-encoded and ETB plasmid-borne [2]. The detection of both toxin genes in our isolate reinforces the potential for more extensive skin involvement and underscores the virulence of ST121 in SSSS contexts. WGS also identified leukocidins, proteases, multiple enterotoxins, and *icaC* gene, indicating high virulence, immune evasion, and biofilm potential. This genomic profile underscores its hypervirulent nature and capacity for persistence in clinical environments. Globally, ST121 has emerged as a dominant cause of exfoliative toxin–mediated disease in children. In Houston, ST121–associated SSSS cases tripled between 2015 and 2018, reflecting rising clinical relevance [18]. Molecular surveys reveal that although ST121 may only represent ~10% of the population-carried *S. aureus*, it accounts for nearly 70% of disease-causing isolates—a disproportion suggestive of virulence-driven selection [3].

The later detection of a distinct ST30 MRSA colonizing strain, lacking exfoliative toxins yet carrying *mecA* and various superantigens (e.g., *TSST-1*), is noteworthy. This finding underscores the dynamic microbial ecology in NICUs, where pathogen replacement can occur after successful treatment of an initial infection. It also emphasizes that colonization by virulent, resistant organisms—while not causing immediate disease—remains a risk for future invasive infection, particularly in medically fragile infants. Presence of virulence factors other than ETs (e.g., TSST-1, enterotoxins, biofilm genes) underscores the need for continued vigilance, despite the absence of skin disease [19]. ST30–MRSA is known as a community-associated lineage that can colonize neonates in NICUs and is implicated in both colonization and outbreaks [20]. MRSA colonization in NICUs is common; meta-analyses report acquisition rates up to 6%, with colonization greatly increasing the risk of subsequent infection [20].

Several prior reports provide valuable diagnostic and therapeutic challenges that parallel and contrast to our case. Hennigan and Riley (2016) reviewed neonatal SSSS cases, emphasizing its rarity but also its potential to recur, particularly when colonization persists [21]. Outbreak investigations highlight the high transmissibility of ET-producing strains among preterm cohorts [3]. Azarian et al. (2021) demonstrated the power of WGS in outbreak settings, tracing ET-positive *S. aureus* strains across patient and healthcare worker populations [2].

A broader retrospective (over 8 years in one center) found SSSS in neonates typically presenting with fever, skin peeling upon pressure (positive Nikolsky), irritability, and refusal to feed—similar to our infant’s symptoms [22]. Systematic reviews emphasize prompt recognition and early antibiotic coverage (e.g., oxacillin) combined with supportive care as the backbone of management [4]. Lee et al. described a congenital presentation of SSSS in a preterm infant manifesting at birth, which progressed to a fatal outcome due to secondary *Candida parapsilosis* fungemia, underscoring the vulnerability of preterm infants to rapid deterioration and opportunistic infections [23]. Similarly, Duijsters et al. reported recurrent SSSS in a very low birth weight infant, with initial misdiagnosis delaying targeted therapy; subsequent confirmation of *S. aureus* producing exfoliative toxin A emphasized the importance of high clinical suspicion and timely microbiological confirmation [7]. These reports align with our case, where prematurity compounded disease severity and highlighted the need for prompt toxin-specific diagnostics.

Outbreak analyses, such as the review by Nusman et al., further emphasize that while SSSS diagnosis is often clinical, adjunctive tools like histology, bacterial cultures, and molecular typing can refine case management, particularly in fragile neonatal intensive care settings [3]. The Irish outbreak reported by Neylon et al. demonstrated that neonatal SSSS can follow minor procedural skin breaches (e.g., intramuscular injections, heel pricks) and that colonization among healthcare staff can contribute to transmission, reinforcing the necessity of strict infection control [24]. Systematic review by Mishra et al., recommend early initiation of beta-lactam antibiotics, reserving MRSA coverage for high-risk settings, and avoiding nephrotoxic drugs in neonates [25].

The application of whole-genome sequencing (WGS) in our case provided high-resolution insight—confirming toxin gene presence, strain lineage, and resistance profiles. Beyond diagnosis, WGS facilitated targeted clinical interventions: prompt catheter removal, optimized antibiotic therapy, and heightened infection control awareness [26]. This mirrors broader trends favoring WGS in NICU surveillance: it allows differentiation between colonization and infection, enables tracing transmission events, and supports tailored decolonization strategies [27].

The management strategy was exemplary: empiric vancomycin was initiated promptly, with a switch to oxacillin once susceptibility results were available, according to pediatric infectious disease guidelines [10]. Removal of the central venous catheter was a decisive intervention, eliminating a probable source of bacteremia and toxin dissemination. The infant’s rapid regression of lesions, weight gain, and stabilization highlight the benefits of early multidisciplinary involvement, combining antimicrobial stewardship, invasive device management, and supportive care. The successful clinical course parallels the generally favorable prognosis in pediatric SSSS when appropriate interventions are timely [10].

## 5. Conclusions

This case highlights the importance of early recognition and prompt management of staphylococcal scalded skin syndrome in preterm neonates, particularly in the context of toxin-producing *S. aureus* strains. The identification of an ST121 isolate carrying both *eta* and *etb* genes through next-generation sequencing underscores the clinical value of genomic tools in guiding targeted therapy and informing infection control measures. Our findings add to the growing body of literature on the molecular epidemiology of SSSS and emphasize the role of genomic surveillance in optimizing neonatal care and preventing outbreaks in vulnerable populations.

## Figures and Tables

**Figure 1 antibiotics-14-01017-f001:**
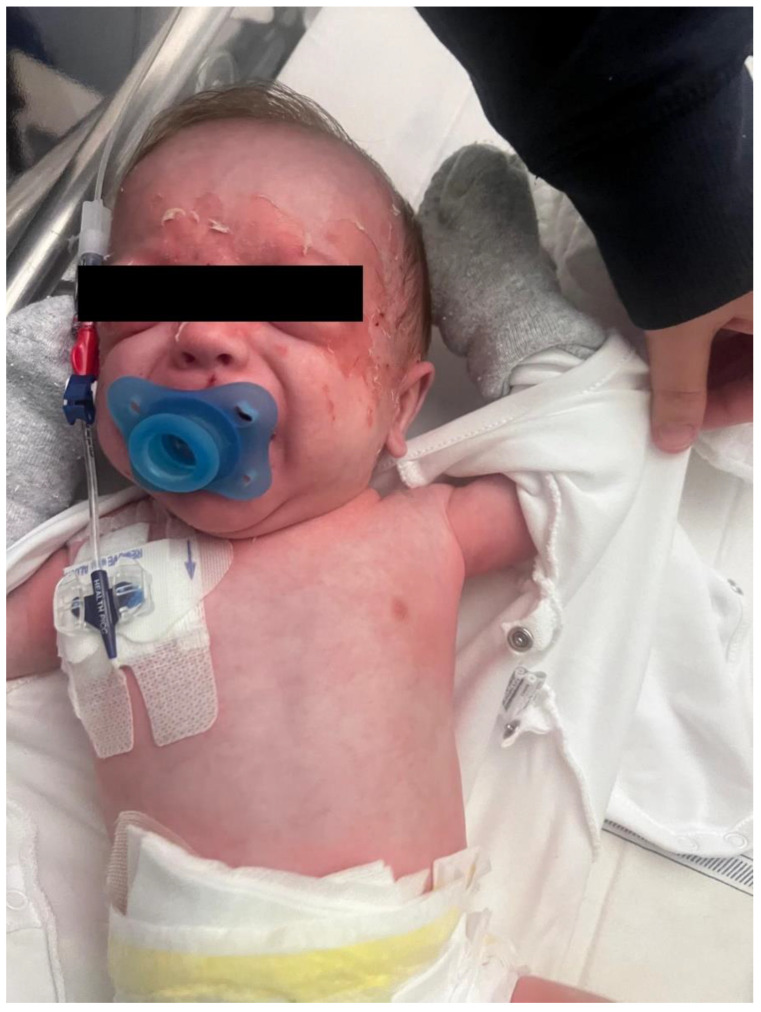
Diffuse erythematous rash with flaccid bullae and periorificial desquamation, especially in the facial area.

**Figure 2 antibiotics-14-01017-f002:**
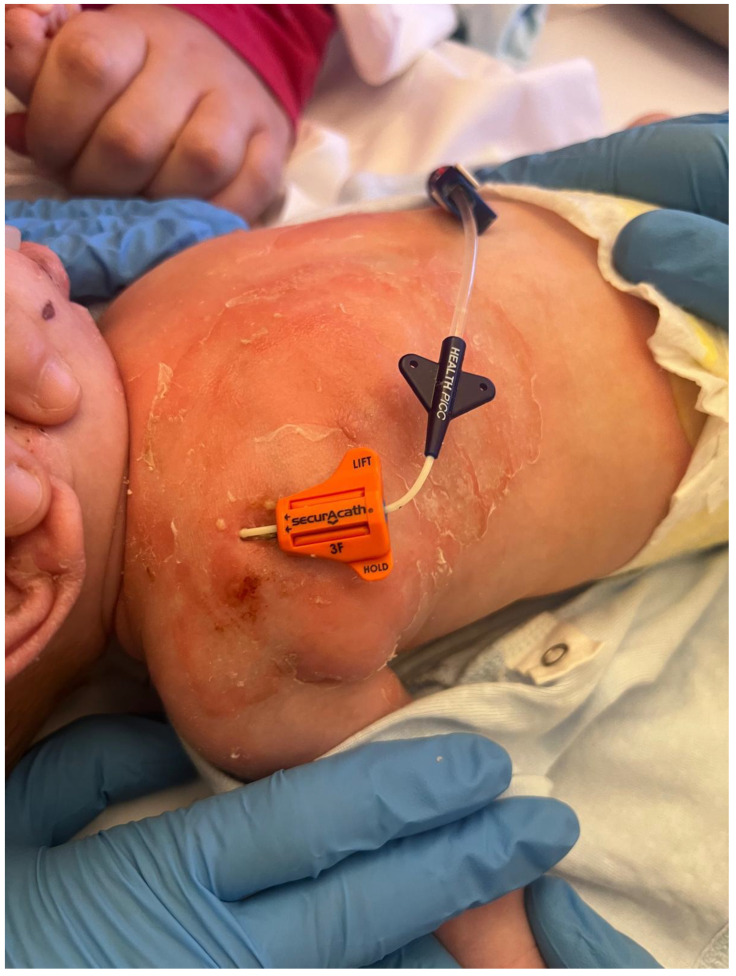
Diffuse erythematous rash with flaccid bullae and desquamation in the CVC insertion area.

**Table 1 antibiotics-14-01017-t001:** Antimicrobial susceptibility profile of *S. aureus* isolate with MIC values and interpretations.

Antimicrobial Agent	MIC (μg/mL)	S/I/R
Amikacin	8	S
Azithromycin	1	S
Ceftaroline	0.25	S
Cirpofloxacin	0.5	I
Clindamycin	≤0.12	S
Daptomycin	0.5	S
Erythromycin	≤0.5	S
Gentamicin	≤1	S
Levofloxacin	≤0.5	I
Linezolid	4	S
Moxifloxacin	≤0.25	S
Oxacillin	0.5	S
Teicoplanin	≤1	S
Tetracycline	≤1	S
Tigecycline	≤0.25	S
Tobramycin	≤1	S
Amikacin	≤1/19	S
Azithromycin	1	S

## Data Availability

The original contributions presented in this study are included in the article material. Further inquiries can be directed to the corresponding author.

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
