# Peer review of "Staphylococcal Scalded Skin Syndrome (SSSS) in a Preterm Infant: Clinical Presentation and Role of Next-Generation Sequencing in Toxin Gene Identification—A Case Report"

_antibiotics, 2025, doi:10.3390/antibiotics14101017_

Round 1
Reviewer 1 Report
Comments and Suggestions for Authors
This is a very interesting article describing a case report of a neonate suffering from Staphylococcal scalded skin syndrome (SSSS). The authors highlight the role of Next-Generation Sequencing (WGS) in microbiological diagnostics. An important question arises: is WGS currently applied in hospital diagnostics, or is it mainly reserved for research purposes? It would be valuable to emphasize the significance of WGS in diagnostics in the conclusion section.
I have one minor comment: in the Results section, it is unclear whether the sentence “Additionally, the strain expresses bicomponent leukocidins LukED and gamma-hemolysin HlgABC, which contribute to immune evasion and cytotoxicity against host leukocytes [9]” refers to the strain isolated from the infant in this study or to a strain described in the literature by F. Joly et al. The same ambiguity appears in lines 153–156.
What is the main question addressed by the research? This case report describes an extremely premature infant with Staphylococcal Scalded Skin Syndrome (SSSS) and emphasizes the advantages of employing WSG as a valuable diagnostic tool.
Do you consider the topic original or relevant to the field? The topic of the article is very important from a medical point of view, as SSSS affects neonates and children younger than six years.
What does it add to the subject area compared with other published material? The importance of WGS in diagnostics is underscored in this case. This method allows for the identification of virulence genes, such as those encoding the exfoliative toxins ETA and ETB, the detection of resistance determinants, and the determination of sequence type (ST). WGS not only confirmed the dermatological diagnosis of SSSS but also supported a therapeutic adjustment. Initial therapy with vancomycin was subsequently replaced with oxacillin, upon confirmation of MSSA.
What specific improvements should the authors consider regarding the methodology? The authors provided complete and accurate information on DNA extraction, library preparation, sequencing, and bioinformatics analysis. Therefore, I have no reservations regarding the description of the methodology. Importantly, the work also includes photographs illustrating SSSS.
Are the conclusions consistent with the evidence and arguments presented and do they address the main question posed? Yes, the authors emphasize the importance of early recognition of Staphylococcal Scalded Skin Syndrome in neonates and demonstrate the clinical value of using next-generation sequencing to guide targeted therapy.
Are the references appropriate? Yes.
Author Response
COMMENTS 1: I have one minor comment: in the Results section, it is unclear whether the sentence “Additionally, the strain expresses bicomponent leukocidins LukED and gamma-hemolysin HlgABC, which contribute to immune evasion and cytotoxicity against host leukocytes [9]” refers to the strain isolated from the infant in this study or to a strain described in the literature by F. Joly et al. The same ambiguity appears in lines 153–156.
Response 1: To address the ambiguity, we have revised the text by clarifying that the genetic determinants encoding the bicomponent leukocidins lukED and hlgABC were identified in the strain analyzed in this study. Furthermore, the descriptive part on their biological role has been maintained, and the citation has been updated with the wording “has been recognized” to indicate that their contribution to immune evasion and cytotoxicity has been established in the literature. This ensures that the reader understands both the genetic finding in our isolate and its biological relevance.
Reviewer 2 Report
Comments and Suggestions for Authors
This manuscript presents the clinical case of an infant with Staphylococcal scalded skin syndrome (SSSS) and the identification of the causative agent using NGS.
The following comments are made:
- Line 53. Italicize scientific names. Review throughout the text.
- Line 70. Indicate that they are genes, or if referring to toxins, capitalize them and not italicize them. In this case, it is best to refer to the toxins.
- Line 98. Indicate what cultures and tests were performed to confirm S. aureus and how you determined it was MSSA.
- Line 99. Could a nasal or throat swab be performed? Explain whether or not and why.
- The caption for Figure 2 is missing. Include it.
- Line 107-108. Enter the doses used and the start and end date of antibiotic administration.
- Lines 153-190. Protein names are not italicized. Correct throughout.
- Line 168. Indicate in the methodology how this test was performed.
- Line 168. Italicize scientific names. Review and correct throughout.
- Line 174. How did you know it was an MRSA strain? Explain.
- Line 220. Indicate the icaC is a gene.
- Line 228. Remove the bold.
- Discuss how the patient became infected. Where did he become infected with the MSSA strain? Indicate whether you looked for the presence of the strain in the NICU.
- Line 301. It is unlikely that the patient gave you authorization for publication and informed consent. Explain this.
- Review and standardize the references; see Instructions for Authors. There is missing data as in references 4, 10, etc.
Author Response
Comment 1. Line 53. Italicize scientific names. Review throughout the text.
Response 1. All scientific names have been carefully reviewed and consistently italicized throughout the entire manuscript in accordance with the standard nomenclature guidelines.
Comment 2. Line 70. Indicate that they are genes, or if referring to toxins, capitalize them and not italicize them. In this case, it is best to refer to the toxins.
Response 2. The text has been revised to clearly differentiate between genes and toxins. In this specific context, the reference has been modified to toxins, with the appropriate capitalization and formatting applied.
Comment 3. Line 98. Indicate what cultures and tests were performed to confirm S. aureus and how you determined it was MSSA.
Response 3. A dedicated paragraph has been added to the Methods section, providing a detailed description of the microbiological cultures and confirmatory tests performed to identify S. aureus and to determine its methicillin-sensitive phenotype (MSSA).
Comment 4. Line 99. Could a nasal or throat swab be performed? Explain whether or not and why.
Response 4. A specific section has been added. Considering that methicillin-sensitive Staphylococcus aureus (MSSA) was already isolated from both blood cultures and auricular swabs, nasal swabbing was not performed. This decision was based on the redundancy of further sampling and the sufficient diagnostic evidence already available.
Comment 5. The caption for Figure 2 is missing. Include it.
Response 5. A complete and appropriate caption for Figure 2 has been included in the revised version of the manuscript.
Comment 6. Line 107-108. Enter the doses used and the start and end date of antibiotic administration.
Response 6. The Methods section has been enriched with detailed therapeutic information: Vancomycin was administered at a dosage of 60 mg/kg/day in four divided doses, starting from the day of clinical onset and continued until phenotypic antibiotic susceptibility results from blood cultures became available. Subsequently, therapy was switched to oxacillin at a dosage of 200 mg/kg/day, also divided into four daily administrations.
Comment 7. Lines 153-190. Protein names are not italicized. Correct throughout.
Response 7. A thorough revision has been carried out. All proteins, toxins, and genes have been consistently formatted according to the appropriate conventions.
Comment 8. Line 168. Indicate in the methodology how this test was performed.
Response 8. An additional methodological description has been inserted, detailing the procedures followed for the referenced test.
Comment 9. Line 168. Italicize scientific names. Review and correct throughout.
Response 9. The manuscript has been meticulously reviewed, and all scientific names, including those at line 168, have been properly italicized.
Comment 10. Line 174. How did you know it was an MRSA strain? Explain.
Response 10. A dedicated section has been added to clarify the diagnostic procedures. Identification as an MRSA strain was based on nasal swab screening combined with microbiological testing.
Comment 11. Line 220. Indicate the icaC is a gene.
Response 11. The text has been corrected to clearly specify that icaC is a gene. This adjustment has also been harmonized with the overall formatting applied to genes, proteins, and toxins.
Comment 12. Line 228. Remove the bold.
Response 12. The unnecessary bold formatting has been removed as requested.
Comment 13. Discuss how the patient became infected. Where did he become infected with the MSSA strain? Indicate whether you looked for the presence of the strain in the NICU.
Response 13. A dedicated paragraph has been incorporated into the Discussion section. These infections have been interpreted as community-acquired, since MSSA strains are not usually circulating within the NICU. Moreover, no other neonates in the unit presented similar symptoms. This supports the hypothesis that the infection originated from the mother. Screening for strain circulation in the NICU was not performed, as this would require whole genome sequencing (WGS)-based surveillance, and outbreak criteria (≥ 2 cases in the same ward) were not met.
Comment 14. Line 301. It is unlikely that the patient gave you authorization for publication and informed consent. Explain this.
Response 14. We thank the Reviewer for pointing this out. The manuscript has been corrected: both parents provided written informed consent for the publication of the case, including images of their child. The study objectives and implications were fully explained to them, and both parents expressed their understanding and signed the consent form.
Comment 15. Review and standardize the references; see Instructions for Authors. There is missing data as in references 4, 10, etc.
Response 15. All references have been carefully reviewed, standardized, and updated in accordance with MDPI guidelines. Missing data in references (e.g., 4 and 10) have been corrected and the overall format has been homogenized.
Round 2
Reviewer 2 Report
Comments and Suggestions for Authors
- Line 101. In SCCmec, "mec" is italicized.
- Lines 98-108. Could you add references to your methodology?
- Lines 187-189. Add references.
- Lines 233, 241, 270. You still haven't italicized the name of the bacteria. Do a thorough review.
- Line 323. The infants couldn't have given informed consent.
- Reference 4 is still missing volume and pages; Reference 23 is missing pages; Reference 25 is repeated in Reference 23. Conduct a thorough review of the References.
Author Response
I checked the manuscript and made a few small changes. The manuscript is ready for me.